# Metal Cluster Triggered-Assembling Heterogeneous Au-Ag Nanoclusters with Highly Loading Performance and Biocompatible Capability

**DOI:** 10.3390/ijms231911197

**Published:** 2022-09-23

**Authors:** Xiaoxiao He, Xiaohong Ma, Yujun Yang, Xi Hu, Teng Wang, Shiyue Chen, Xiang Mao

**Affiliations:** 1State Key Laboratory of Ultrasound in Medicine and Engineering, College of Biomedical Engineering, Chongqing Medical University, Chongqing 400016, China; 2Chongqing Key Laboratory of Biomedical Engineering, College of Biomedical Engineering, Chongqing Medical University, Chongqing 400016, China; 3State Key Laboratory of Multi-Phase Complex Systems, Institute of Process Engineering, Chinese Academy of Sciences, Beijing 100190, China; 4Key Laboratory of Laboratory Medical Diagnostics, Ministry of Education, Department of Laboratory Medicine, Chongqing Medical Laboratory Microfluidics and SPRi Engineering Research Center, Chongqing Medical University, Chongqing 400016, China

**Keywords:** Au-Ag NCs, heterogeneous structures, assembling, drug carriers, biocompatibility

## Abstract

In this work, we firstly report the preparation of heterogeneously assembled structures Au-Ag nanoclusters (NCs) as good drug carriers with high loading performance and biocompatible capability. As glutathione-protected Au and Ag clusters self-assembled into porous Au-Ag NCs, the size value is about 1.358 (±0.05) nm. The morphology characterization revealed that the diameter of Au-Ag NCs is approximately 120 nm, as well as the corresponding potential ability in loading performance of the metal cluster triggered-assembling process. Compared with individual components, the stability and loading performance of heterogeneous Au-Ag NCs were improved and exhibit that the relative biocompatibility was enhanced. The exact information about this is that cell viability was approximately to 98% when cells were incubated with 100 µg mL^−1^ particle solution for 3 days. The drug release of Adriamycin from Au-Ag NCs was carried out in PBS at pH = 7.4 and 5.8, respectively. By simulating in vivo and tumor microenvironment, the release efficiency could reach over 65% at pH = 5.8 but less than 30% at pH = 7.2. Using an ultrasound field as external environment can accelerate the assembling process while metal clusters triggered assembling Au-Ag NCs. The size and morphology of the assembled Au-Ag NCs can be controlled by using different power parameters (8 W, 13 W, 18 W) under ambient atmosphere. Overall, a novel approach is exhibited, which conveys assembling work for metal clusters triggers into heterogeneous structures with porous characteristic. Its existing properties such as water-solubility, stability, low toxicity and capsulation can be considered as dependable agents in various biomedical applications and drug carriers in immunotherapies.

## 1. Introduction

Metal nanomaterials (NMs) are widely used in the fields of catalysis, sensing, and biomedicine because of their special physicochemical properties and special constructional characteristics [1,2,3,4,5]. In addition, different types of metal NMs have distinct optical, electrical, and catalytic properties [6,7,8]. Sorted by crystalline features, metal NMs could be classified into representative references, based on pure-phased crystalline such as nanoparticles (NPs) with highly crystalline properties and metal clusters formed by atoms aggregations [9]. When their properties were evaluated in physicochemical works, metal clusters drew the most interest among composite materials due to their unique characteristics [10,11,12,13]. Nanoclusters consisting of fewer atoms have a higher specific surface area, which might perform the promised function actively. Obviously, it is feasible to expand the possibility by integrating different metal clusters or by changing the composition ratio of metal atoms in clusters by precisely controlling the molar amount of metal elements [14,15,16,17]. In this case, various metal-cluster based materials with different physicochemical properties can be made from one pure-phased metallic element by adjusting the number of constituent atoms of metallic nanoclusters (NCs) [18,19]. When multiple elements are used, it is possible to further increase the functional diversity [20,21]. Similarly, the precious metal clusters are always composed of several hundreds of atoms, such as gold (Au), silver (Ag), and copper (Cu), which have attracted significant attention due to their unique properties and related applications [22,23,24,25]. In addition, the noble metal clusters have good fluorescence property [26,27,28,29] due to the spatial confinement of free electrons, which leads to discrete electron hopping [30,31], with the electron energy levels dispersing from continuous to discontinuous states. The electron hopping becomes active, resulting in strong light absorption [32,33,34]. Potentially, the allowed molecular-like properties could arise from NCs’ size or morphological formation, which would imply interesting optical property, structural activity, and physical or chemical catalysis [35,36]. It enhances the previous illustration of functional materials and shows that precious metal NCs have a wide range of promising applications.

Extended works on noble metal NCs have been widely applied as innovative fluorescent NMs in biological applications, environmental sciences, pharmacy, and other disciplines [26,37,38,39,40]. Studies have showed that Au NCs possess a killing effect on both Gram-positive and Gram-negative bacteria. The interaction of Au NCs with bacteria may lead to an increase in intracellular reactive oxygen species production in order to kill bacteria [41]. Ag NCs have antimicrobial effects and are suitable for topical antimicrobial delivery [42]. Although some progress has been made, there are still some problems and challenges, such as the small particle size of metal NCs, where it is difficult to control the reactivity, harsh reaction conditions, unclear surface, interfacial structures, and low quantum yields [43,44,45]. Due to the spectral activity and surface activity of functional surfactants, it could provide functional groups (-NH_2_, -COOH, -SH) and imply an assembly mechanism in forming superstructures. It could be proposed that the assembled structures would be more stable and widely used by comparing it with individual cluster. Shi et al. have successfully synthesized chiral Au NCs assemblies with strong circular dichroism (CD) by using atomically precise Au NCs to fabricate ordered assembled structures with emerging optical activity [46]. By understanding the mechanism of assembly structure, noble metal NCs would be excellent candidates for assembled building blocks [47,48]. Due to their thermodynamic stability, the number of constituent atoms of various types can be easily synthesized in large quantities and could assemble heterogeneous structure at the atomic level [49]. Yahia et al. reported that cationic polymers mediated the self-assembly of Au NCs into Au NPs of about 120 nm in diameter while the aggregation-induced emission phenomenon (AIE) resulted in a 4-fold enhancement of luminescence due to electrostatic interactions between the polyelectrolyte and the stable surface ligands of Au cluster [50]. The previous works exhibit that there is difficulty in assembling small sized NCs to artificially control a nanosphere or nanocage structure. Chakraborty et al. suggested atomically accurate, self-assembled structures (Au cluster) that can enclose plasmonic Au nanorods and also enhanced the detection limits of sensor devices [51]. Yonesato et al. presented a controlled assembly method for Ag construction rather than signal cluster [52]. However, the assembled structures were mainly classified as homogeneous Au-Au and Ag-Ag formations [15,53,54]. These homogeneous NCs gives remarkable properties through adjusting organic ligands, which implied the simple assembly modes and controllable reaction processes. However, the challenge exists beyond the limitations, poor properties, and weak structural formation because of its single consistence of homogeneous NCs. It seldom leads to low sorts of assembled novel structures and interested physical and chemical properties, which is different from those of individual cluster [15,55].

Herein, we present heterogeneous assembled structures of Au-Ag by metal cluster triggered by the assembling process in aqueous medium. It performed well in a drug loading capacity and biocompatibility due to the porous characterization, which not only retained the properties of individual cluster (Au, Ag), but also achieved new properties and extended the application fields. During the assembling process, the resultant Au-Ag NCs demonstrates glutathione (GSH) promotes combination or integration between Au and Ag cluster. Au-Ag heterogenous NCs are well-fabricated, the morphology is adjusted by integrating with previous polymers (poly-vinylpyrrolidone, polyethylene glycol400 and poly-ethyleneimine). Adding an ultrasound field implies that this kind of physical field can induce the growth of the assembled NCs in assembling processes. The functional groups on its surface are mainly amino-NH_2_ and carboxyl-COOH, which are hydrophilic molecules with high biocompatibility (Figure 1a).

## 2. Result and Discussion

### 2.1. Synthesis and Characterization of Assembled Heterogeneous Au-Ag NCs

Heterogeneous Au-Ag NCs were fabricated during in-assembly approaches via a simple approach, as shown in Figure 1a; the scheme could reflect the main bond-forming process in a chemical environment (Figure 1c). A similar phenomenon was already conveyed exactly in our previous works [56]. In the processes of assembled heterogeneous Au-Ag NCs, the individual Au or Ag cluster were modified by GSH molecule completely, which played a key role in chemical and physical stabilities. As the protector for individual cluster, GSH could apply the different functional groups as link agents while meeting other coupled groups for constructing amid functions. Moreover, there have been many findings which suggest that assembled nanostructures are composed through hydrogen bonds, electrostatic and Vander Waals interactions as much as possible. Similarly, the assembled behaviors might provide a large number of precise, ingenious and accurate for different structures. The linkages were realized through bonding of approaching surfactants, which was well utilized in the synthesis process; moreover, its molecular functional groups such as -COOH (carboxyl) and -NH_2_ (amino) provided a high possibility for the assembling processes (Figure 1c). The morphology and size of the formed Au-Ag NCs changed considerably when compared to the assembled heterogeneous Au-Ag NCs (Appendix A) for single Au or Ag clusters (Appendix A). Compared to the monolithic nanoclusters, the particle size of Au-Ag NCs is increased by about 10 times, and the morphology is regular spherical aggregates. Polymers such as PVP and PEG400 were used to enhance the stability of the structures, which could facilitate the growth of the assembled structures throughout the growth process. PEI can be used as a binder to facilitate the generation of the assembled structure. As shown in Appendix A, in the growth process of assembled heterogeneous Au-Ag NCs, individual nanoclusters gradually aggregate from small irregular agglomerated states to regular spherical aggregates under the influence of polymer materials, hydrogen bonding, and electrostatic interactions. In this work, the bonding response and electrostatic adsorptions, and the furtherance in assembling would lead to obtaining the heterogeneous structures, but also the diameter is approximately two hundred nanometers larger. The modification of assembled structures can not only ensure the dispersion and stability but also promote the circulation time in vivo and reduce the scavenging effect. Furthermore, these polymers could improve the efficiency of metal clusters boning together through synergistic interactions. It can be treated as one soft-template with different clusters implanted on its surface. It enhanced Au and Ag clusters assembled efficiently and conveyed regular morphology or structural formations. It was especially good for making heterogeneous Au-Ag NCs, such as bridge-linkage between Au and Ag clusters, which were obtained and modified completely by polymers. HR-TEM characterizations proved internal information and, additionally, the morphological shapes appeared as regular spheres (Figure 1b,d). It demonstrated the Au-Ag cluster @PVP and Au-Ag cluster @PEG 400 heterogeneously assembled structures at RT with an average diameter of 100 nm. It also implied that the pores appear in the assembled structure via HR-TEM characterization. All of these assembled heterogeneous structures eventually showed morphological size uniformity and porous characteristics. By adjusting the added mole quantification of each component, the high resolution of analysis should be addressed for understanding structural information.

As shown in Figure 2a,c, the TEM images conveyed Au and Ag NCs could be integrated together as sphere appearance by using PVP as soft template in assembling processes. The HR-TEM characterization obviously embodied the structural particularity, it indicated the pore appeared (Figure 2b) and metal cluster could link together after using polymer molecule (PVP) as synergistic reaction. It revealed one principle about the assembled structure but also reflected the uniformity of the homogeneous pore well. Additionally, the observed pore should be adjusted in the assembling process. In detail, the internal structural measurement to detailed analysis of these assembled Au-Ag NCs was achieved (Figure 2c), it implied the possible mechanism in fabricating heterogeneity was mainly determined by successful integration between different components. The characterization of elemental (Au, Ag) composition and distribution could prove the assembled process; furthermore, it certified the homogenous consist status while Au integrating with Ag clusters. In addition, the structural morphology was also confirmed by the distribution of Au and Ag as much as possible. It confirmed that the Au (yellow) and Ag (green) are spherically and uniformly distributed in the EDS mapping results, which proved the elemental composition of this heterogeneous Au-Ag NCs. It indicated the distribution of Au and Ag in the assembled structure as a homogeneous distribution of porous spherical structure with porous characteristic (Figure 3a). After characterizations, the pore size can reach 3.358 nm, which was just a pure phased metal cluster with different polymers. In these procedures, metal clusters were assembled together by the condensation reaction of amino and carboxyl groups, but also the hydrogen bonding and electrostatic interactions attended timely. The polymer materials (PVP, PEG400 and PEI) have the stability to remain as dispersed and non-agglomerated as possible when integrated with individual metal clusters. The linkage in this kind of building blocks the enhanced assembling processes; however, the soft template also provided a precondition in order to construct regular morphology or keep the uniformity in fabricating Au-Ag NCs. In all this, the assembled Au-Ag structure should be considered as a spontaneous fabrication process, with hydrogen bond, van der Waals force and Ionic bond synergism as the main driving force.

Fourier transform infrared spectroscopy (FT-IR) was used to demonstrate the presence of different static functional groups in assembling works. It could convey the initial linkage between each component and chemical reaction completely. The pure glutathione (GSH) and PVP could imply the obvious references while comparing with the resultant Au-Ag cluster @PVP NCs. As shown in Figure 3b, the vibrational spectra of alkyl C-H stretching in the assembled Au-Ag cluster @PVP was characteristic at 2956 cm^−1^. The whole reaction is in the aqueous phase at room temperature, which will not damage the structure of PVP itself, and PVP acts as a soft stencil and binder in the whole reaction process, and PVP is a water-soluble polymer with good dispersion to prevent particles from gathering and precipitating each other. The spectrum of Au-Ag cluster @PVP NCs comparable to the one of only PVP, indicating that PVP can be used as a stabilizer. The observed 1001 cm^−1^ to 1058 cm^−1^ peaks can be attributed to C-O-C stretching, while 1492 cm^−1^ was from C-N stretching. The absorption peaks near 3415 cm^−1^ are O-H stretching vibrations, near 1543 cm^−1^ are N-H stretching vibrations, and the bond at 1058 cm^−1^ is due to C-O stretching vibrations [57], C=O has a characteristic absorption peak near 1680 cm^−1^. The thiol group (-SH) did not appear at 2520 cm^−1^ due to the integration of GSH and metal clusters (Au and Ag). During the assembled process, the main bonding works were realized for forming amid linkage. The hydrophilic functional groups (carboxyl, -COOH; amino, -NH_2_) could couple together in the form of amid linkages, which was enhanced by using polymeric materials. Therefore, the changes in peak position and relative intensity might be considered as one essential factor in assembling work. Meanwhile, the Au-Ag cluster @PEG400 showed the vibration spectra of PEG, Au, and Ag cluster observed that the FI-IR absorbing functional groups mainly have C-C stretching vibrations near 1358 cm^−1^, CH_2_ stretching vibrations at 1342 cm^−1^, and C-O stretching vibrations at 1000–1160 cm^−1^, respectively (Figure 3b). The combination of PEG and Au-Ag NCs reflected -OH stretching vibration (2110 cm^−1^), and -OH bending vibration mode (1650 cm^−1^). Moreover, the absorption peak of -OH was at 3500 to 3200 cm^−1^ [24]. By using PEI as a synergistic agent in fabricating Au-Ag NCs, there are the same influences and appearance of functional groups (Appendix A). Furthermore, the relative stability of these assembled structures (Au-Ag cluster @PVP, Au-Ag cluster @PEG400) can be realized by measuring their surface charge in water media and buffer solutions. As shown in Figure 3c and Appendix A, zeta potential indicated these assembled structures have a negative surface charge, while PVP and PEG400 prevent the aggregation of particles through spatial site resistance, forming a stable colloidal structure that can maintain its stability in DI water and PBS solutions. However, the shielding of the surface charge of the nanoclusters by the polymer material leads to an increase in the zeta potential value. The difference in zeta potential value between two kinds of assembled structure can be ascribed to the chemical charge adsorption through particular chemical groups (-C-OH, -C=O) influence. Additionally, it proved the structural stability and water solubility.

### 2.2. Drug Loading and Releasing Performance of Assembled Au-Ag NCs

Due to the porous characteristic, this assembled Au-Ag NCs can be used as drug carriers in further investigations. As shown in Figure 4a,b, it conveyed the characteristic peak near 480 nm and 590 nm after the Au-Ag cluster@PEG400 loaded DOX molecule, it illustrated the emission spectrum difference before and after DOX molecule loaded completely. Among them, the assembled Au-Ag cluster@PEG400 had no absorption peak near 480 nm, which confirmed that DOX could be loaded successfully on Au-Ag cluster@PEG400. After loaded DOX, the TEM image of whole construction was shown in Figure 4c, it exhibited a similar morphology to the original structures due to the porous loading particularity. The further investigation of combination could be attributed to electron adsorption, pore adsorption and molecular hydrogen bonding [58,59]. The exhibition in loading performance could be well-conveyed in optical measurement, and it gave the absorbance peaks status. Appendix A showed that DOX is interconnected with the assembled heterogeneous Au-Ag NCs, in which DOX was encapsulated in the assembled structure, which might lead to an increase in particle size. Similarly, the typical Au-Ag cluster @PVP NCs were also checked for measuring its loading performance, as shown in Figure 4d,e. It demonstrated the absorption peaks (480 nm) and emission difference (590 nm), which proved loading performance directly. Figure 4f and Appendix A showed TEM images, showing that the surface of the assembled structure loaded drug has a polymer film with increased particle size. Furthermore, the morphological exhibition changed more obviously than PEG400 modified Au-Ag NCs when loading the DOX molecule. This can be attributed to the target molecules might enhance the hindrance while integrating with chemical electronegativity groups.

In another word, the drug entered the assembled structure and loaded in the porous structure by π-π stacking, molecular hydrogen bonding and electrostatic adsorption during in loading process. Here, PVP might be thought of as a soft template but also possessed a certain adsorption effect, so that DOX can be successfully loaded on the assembled Au-Ag cluster @PVP. Au and Ag cluster with particle size of less than 10 nm were assembled by PEI into NCs with size of around 50 nm (Appendix A). In addition, the loading performance of the heterogeneous Au-Ag cluster @PEI was also characterized as shown in Appendix A. There are special phenomena such as color changes in the merging processes and the particularity of red-shift and bonding formations. When added DOX, the color changed from orange-red to blue-purple, and the UV absorption peak of the Au-Ag cluster @PEI-DOX shifted from 480 nm to 580 nm, which was due to the formation of hydrogen bond between the carbonyl group of DOX (as hydrogen acceptor) and the -NH_2_ group of PEI (hydrogen donor) [60]. Amino groups interacted rapidly with the carbonyl groups on DOX to form hydrogen bonds in aqueous solution, and its morphological features can be seen in the TEM images as a striated structure with the appearance of a lattice (Appendix A). All of the above loading performances implied the feasibility of treating these assembled Au-Ag NCs as one acceptable carrier in further drug delivery. The estimation of the release performance was realized in vitro DOX release evaluation of Au-Ag cluster @PVP-DOX and Au-Ag cluster@PEG400-DOX at RT in different pH solutions (PBS pH = 7.4 and 5.8). The qualitative analysis of the UV-vis absorption spectra as in Figure 5a,b, demonstrated that DOX can be released significantly more with DOX released at pH = 5.8 than at pH = 7.4 in the form of higher absorbance of DOX (480 nm). Similarly, the fluorescence spectra also show a higher absorbance of DOX at pH 5.8 at 590 nm and a higher release of DOX (Figure 5c,d). It demonstrated DOX was released more efficiently in an acidic environment as measured in optical property characterizations. It fully conveyed that polymers (PEI, PVP, PEG400) could assist the assembling processes but also illustrated that heterogeneous Au-Ag NCs has promising applications in the form of biological carriers for targeted tumor therapy.

In order to evaluate the releasing performance, the whole assembled NCs were cultivated in PBS solution (pH = 5.8). It could reach 69.8% (Au-Ag cluster @PVP) and 65.8% (Au-Ag cluster @PEG400) of DOX releasing efficiency within 48 h. In this process, DOX is connected to Au-Ag cluster @PVP by π-π stacking and is also integrated during molecular hydrogen bonding and electrostatic adsorption. An amid group could be achieved by Au-Ag cluster@PEG400 integrated with DOX, in which case it might form electronic interactions, hydrazone bonding, and amide bonds. As shown in Appendix A, the standard curve was used to quantify the concentration of DOX through UV-vis characterizations. The loading capacity of the Au-Ag cluster @PVP and the Au-Ag cluster @PEG400 could reach 5.8% and 13.1%, respectively (Figure 6a). Due to the lower relative molecular mass of PEG400, this spatial repulsion was relatively weak, and the more clusters could attach to each other, the higher adhesion it surely achieved than PVP [61]. Polymers on the surface of assembled Au-Au NCs acted as a colloidal stabilizer through spatial repulsion. Furthermore, the drug loading rate of Au-Ag cluster@PEG400 is higher than that of Au-Ag cluster @PVP because the affinity of PEG400 was stronger than that of PVP.

By using lyophilization, it found that the Au-Ag cluster @PEG400 could not be lyophilized to a solid state, which indicated that its adhesion was stronger than PVP and loaded more drugs. As shown in Figure 6b, the release performance was estimated in different pH conditions, which indicated that the higher releasing efficiency could be achieved in pH = 5.8 solvent. The Au-Ag cluster @PVP and Au-Ag cluster @PEG400 can imply that 62% and 72% of DOX is released within 12 h, respectively. It was significantly faster at pH = 5.8 compared to pH = 7.2. It underwent a buffer solution (PBS, pH = 7.2), which meant about 19% and 23% release efficiency of DOX from these two Au-Ag cluster @PVP and Au-Ag cluster@PEG400, respectively. In weak acidic condition, the protonation of DOX in the amine group increases its solubility, which enhances the release of Au-Ag cluster @PVP-DOX and Au-Ag cluster@PEG400-DOX [62]. In addition, these different types of assembled Au-Ag NCs not only lead to improved physical properties, enhanced enrichment function, and increased drug loading rate, but also met the requirements for degradation and self-clearance in living organisms [63,64]. It indicated that the assembled structure would accelerate the release performance in tumor cells due to the tumor and lysosomes’ acidic micro-environment. It could effectively reduce the toxic side effects of DOX to enhance the tumor killing effect.

### 2.3. Biocompatibility of Assembled Heterogeneous Au-Ag NCs

The biocompatibility was evaluated through heterogeneous Au-Ag NCs integrated with cell lines (239T, HACAT, IMEF) at different culture times and concentrations. Live cells were stained using CA and dead cells were stained with PI. Different fluorescence and digital microscopy images of different cell lines were cultured for 2 days, as shown in Figure 7a and Appendix A, respectively. The cell activity of the Au-Ag cluster @PVP, Au-Ag cluster@PEG400, and Au-Ag cluster @PEI groups exceeded 90% within 2 days. It can be visually observed in the fluorogram of cell distribution (green fluorescence without red fluorescence). The results of live-dead cell staining showed that the assembled heterogeneous Au-Ag NCs is highly biocompatible and does not affect cell growth and reproduction with less cytotoxicity. Cell viability in the presence of Au-Ag cluster @PVP showed 100%, 94% and 97% viability at concentrations of 25, 50, and 100 µg mL^−1^, respectively. Even after 2 days, there was no significant change in cell viability, still showed high biocompatibility. The cell viability in the presence of assembled heterogeneous Au-Ag NCs exhibited ≈98%, 95%, and 90% viability at concentrations of 25, 50, and 100 µg mL^−1^, respectively. Cell viability showed high activity after 2 days of increasing concentration incubation. By keeping constant conditions, it showed excellent adaptation to cell types via culture with Au-Ag cluster@PEG400. Similarly, the cell viability of the Au-Ag cluster @PEI was over 90% (Figure 7b). All sets of experimental results showed that the assembled heterogeneous Au-Ag NCs are highly biocompatible.

### 2.4. Synthesis and Characterization of Ultrasonically Induced Assembly of Heterogeneous Au-Ag NCs

The high frequency ultrasonic experiment (HFUE) instrument (6 MHz) was selected in this work. As shown in Appendix A, it is the main work module and circuit diagram of device, which was mainly composed of 4 parts of ultrasonic emission circuit. The ultrasonic field was used to investigate the effects of different powers on the formation of the assembled structure (Figure 8a). After the physical treatment, Au and Ag clusters were induced on the ultrasonic planar coupler to form the assembled structure. Ultrasound fields could accelerate the assemble rate, provide reaction conditions, and promote heterogeneous structures forming processes (Appendix A). By assuming that the ultrasonic waves were emitted from one side, clusters would move toward the acoustic pressure node (red dot in Figure 8b) stop working for 5 s after 10 s of emission. The acoustic cavitation can provide sufficient energy for the initiation of free radical reactions, changing into reactive H· and OH·. As shown in Appendix A, once a certain amplitude of acoustic rarefaction passes through solution, the bubbles rupture violently, releasing the high temperature induced linkage between metal clusters. Ultrasonic irradiation can induce free radicals of OH· and H· radicals (Equation (1)). These radicals recombined to return to their original form or combine to generate H_2_O_2_ and H_2_ (Equations (2) and (3)), and the oxidants and reductants were employed during different reactions of the sonochemical exhibitions. The functional groups such as -NH_2_, -OH and -COOH will fabricated linkage in the form of hydrogen bonds and chemical bonding between Au and Ag NCs. -OH and -COOH can form hydrogen bonds with water molecules in water, and because the electron-giving ability of nitrogen atoms is greater than that of oxygen atoms (-NH_2_ is more electrophilic than -OH), -NH_2_ and -COOH can amide to form peptide bonds(-NH-CO-) (Equation (4)). These chemical bonds connect Au NCs and Ag NCs on the surface of the cluster.
(1)$$H_{2}O\to H\cdot +\mathrm{OH}\cdot$$
(2)$$\mathrm{OH}+\mathrm{OH}\to H_{2}O_{2}$$
(3)$$H\cdot +H\cdot \to H_{2}$$
(4)$${\text{-}\mathrm{NH}}_{2}+\mathrm{COOH}\to {\text{-}\mathrm{NH}\text{-}\mathrm{CO}\text{-}+H}_{2}O$$

Adjusting the parameters, it was found that Au and Ag could incubate with polymer materials completely; this resulted in heterogeneous Au-Ag NCs under different ultrasonic powers of 18 W, 13 W, and 8 W, respectively. It implied that the assembled Au-Ag NCs could appear while using ultrasonic power of 18 W and 8 W (Figure 8c,d). HR-TEM characterization showed a regular spherical construction and a regular pore presence (Appendix A). In terms of PVP, the spherical-like aggregates also appeared while ultrasonic power was reduced to 13 W and 8 W (Appendix A). The resultant structures became larger in particle size and irregular in morphology, as evidenced in Appendix A. Along with ultrasound power increasing, the resultant heterogeneous Au-Ag NCs became smaller and the shape was gradually more regular than before. When PVP containing non-breakable C=O was replaced with PEG400 containing C-O-H and C-O bonds, the formation of the assembled structure was different under the action of ultrasound. The irregular aggregates appeared while ultrasonic power was increase to 13 W and 18 W (Appendix A). The resultant structures became irregular in morphology, as evidenced in Appendix A. Along with ultrasound power increasing, the shape of heterogeneous Au-Ag NCs was gradually more irregular than before. From this works, it can be seen that the addition of different polymeric materials resulted in different assembly processes at the same power. The ultrasound field enhanced the emulsification of non-ionic polymers [65,66,67] but also strengthens the binding between Au and Ag clusters.

## 3. Materials and Methods

### 3.1. Materials

Chloroauric acid (HAuCl_4_, 99%), sodium borohydride (NaBH_4_, 95%), L-glutathione (L-GSH, 99%), polyvinyl pyrrolidone (PVP, Mw: 58,000, 95%), Macrogol 400 (PEG400, Mw: 400, 90%), Polyethyleneimine (PEI, Mw: 800, 90%), Doxo-rubicin Hydrochloride (DOX, 98%), Phosphate buffer (PBS pH = 7.2 and 5.8), Calce-in-AM (CA), propidium iodide (PI) and Cell Counting Kit-8 (CCK8) were supplied by Aladdin Reagent Co., Ltd. (Shanghai, China), Silver nitrate (AgNO_3_, 99%) were provided by KESHI (Chengdu, China). All chemicals were used without further purification.

### 3.2. The Fabrication of Assembled Au-Ag NCs

#### 3.2.1. Synthesis of Au and Ag Clusters

The Au cluster was synthesized based on previous work [41]. 0.0461 g of L-GSH was fully dissolved in 50 mL of aqueous solution, then 0.394 mL of the HAuCl_4_·3H_2_O (0.1 mg mL^−1^) solution was added into the L-GSH aqueous solution. The mixed solution was milky white and was transferred to a magnetic agitator and stirred continuously (500 rpm) for 5 h at 80 °C. The Au clusters was successfully achieved once the color changed to bright yellow. Dialysis purification process was used to depurate the Au cluster. Ag clusters have been previously synthesized [68]. Preparation using the same conditions but replacing the Ag precursor reactant was performed. 0.3073 g of L-GSH was fully dissolved in 50 mL aqueous solution, then 0.0425 g of AgNO_3_ was added into the L-GSH aqueous solution. After 30 min of reaction, 0.0945 g of NaBH_4_ was added to the mixed solution. The mixed solution was milky white and transferred to a magnetic agitator and stirred continuously (500 rpm) for 3 h at −4 °C. The Ag clusters was successfully achieved once the color changed to brownish-black.

#### 3.2.2. Synthesis of Assembling Heterogeneous Au-Ag NCs (Au-Ag Cluster @PVP, Au-Ag Cluster @PEG400 and Au-Ag Cluster @PEI)

The prepared Au NCs (25 mg mL^−1^) and Ag NCs (18 mg mL^−1^) were dialyzed and purified. A total of 7 mL of Au NCs and Ag NCs was transferred into a small beaker and 1 g of PVP was added. The whole mixture was kept at 37 °C for 24 h with continuous stirring, avoiding light during the whole experiment. Similar to the above preparation, Au-Ag cluster @PEG400 and Au-Ag cluster @PEI are formed as Au-Ag cluster @PVP structure.

### 3.3. Drug Loading and Releasing Characteristics Measurements

A total of 2 mL of Doxorubicin hydrochloride water solution (10 mg mL^−1^) was added into 5 mL of Au-Ag cluster @PVP and Au-Ag cluster@PEG400 solution. The solution was continuously stirred in the dark for 2 h. Subsequently, the solution was dialyzed to remove the excessive DOX attached on the surface of the particles. The DOX loading ratio was estimated by the UV-vis spectrometer. To study drug release behaviors, 1 mL of Au-Ag cluster @PVP-DOX or Au-Ag cluster @PEG400-DOX solution were placed in dialysis bag (MW: 3500) and immersed in 50 mL of PBS buffer solution with different pH values of 5.8 and 7.4, the solution was then placed in a shaking incubator (37 °C, 150 rpm), and at determined time period, 1 mL of DOX released medium was sampled, and an equal volume of fresh PBS medium was added to maintain the sink conditions. Then, the released DOX amount was measured by UV-vis spectrometry.

### 3.4. Biocompatibility Measurement and Related Experiments

To investigate the cytotoxic effects of different concentrations of heterogeneous assembly materials on 239T, HACAT, and IMEF cells, the three cells were inoculated in 96-well plates at a density of 5.0 × 10^3^ cells well for 24 h. Media containing Au-Ag cluster @PVP, Au-Ag cluster@PEG400 and Au-Ag cluster @PEI solutions at different concentrations (2, 5, 12.5, 25, 50, 100 µg mL^−1^) were added to 96-well plates, and these cells were continuously cultured for 1 and 2 days. Cellular activity was then assessed by CCK8 according to the manufacturer’s instructions, and cell viability values were measured by measuring absorbance with an enzyme marker. To further quantify cytotoxicity, live dead cell staining experiments were also performed.

### 3.5. Heterogeneous Au-Ag NCs via Assembly Process with Ultrasound Field Function

The selected ultrasonic frequency was 6 MHz, and the ultrasonic induction reaction was performed with a power of 8 W, 13 W, and 18 W, respectively. A total of 3 mL of each prepared gold nanocluster and silver nanocluster solution was added to a small beaker, 0.1 g of PVP or 1 mL of PEG400 was added, and the surface of the ultrasonic transducer was coated with coupling agent to avoid the presence of air to attenuate the ultrasound, and finally, the beaker was placed on the transducer and reacted under the action of ultrasound for 8 h.

### 3.6. Characterization of Heterogeneous NCs

UV-vis absorption spectra were recorded in the range of 200–800 nm by using a Varian Cary 50 UV-Vis Spectrophotometer in absorbance mode. Photoluminescence (PL) spectra were obtained by a Cary Eclipse spectrofluorometer equipped with a xenon lamp. A transmission electron microscope (TEM, JEM-2800, JEOL Ltd., Tokyo, Japan) with an energy dispersive spectrophotometer (TECNAI G2 F20, ER-C, Jülich, Germany) was employed for TEM characterization with an accelerating voltage of 200 kV and a GatanSC200 CCD camera. Automatic specific surface and porosity analyzer BET (ASAP 2460 3.01) are used for average pore size measurements. FT-IR spectra data were recorded with a Nicolet is50 spectrometer (Thermo Fisher, Waltham, MA, USA) in the range from 1500 cm^−1^ to 400 cm^−1^. Zeta potential analyzer (Malvern Zeta sizer Nano ZS90, Malvern, UK) is used to characterize the positive and negative properties of NCs.

## 4. Conclusions

In summary, one effective approach was designed for making heterogeneous Au-Ag NCs. Through the chemical reduction and hybrid methods, it overcomes the drawbacks in preparing heterogeneous structures, which was achieved under organic solvents and heating high temperature. It presented one practical method for the synthesis of metal cluster based heterogeneous constructions. Metal clusters (Au, Ag clusters) can integrate together in the form of regular morphology through synergistic reaction. This could be realized by linkage from chemical bonding and polymer induction in assembling processes. The formed structure exhibits relative structural stability, porous characteristics, and pH dependent properties. The drug loading and releasing performance embodied its structural particularity. It proved that this heterogeneous structure not only retained the functional properties of individual building blocks, but also can bring relative novel functions via assembling procedure. Its biocompatibility exhibition seems to indicate that this kind of heterogeneous NCs could be considered as one carrier in further investigations due to its porous property. The resultant Au-Ag NCs showed that the particle size and structure of assembled heterogeneous Au-Ag NCs were greatly affected with the increase in ultrasound power, and the formation time of assembled structures was shortened at the same time. It proved that physical energy (by acoustic cavitation) releasing might induce the assembling process through influencing the internal bond of polymers. It demonstrated that the design and application of cluster-assembled structures will allow for a new area in the design of novel porous structure and super functional structures, which can be utilized in different applications.

## Figures and Tables

**Figure 1 ijms-23-11197-f001:**
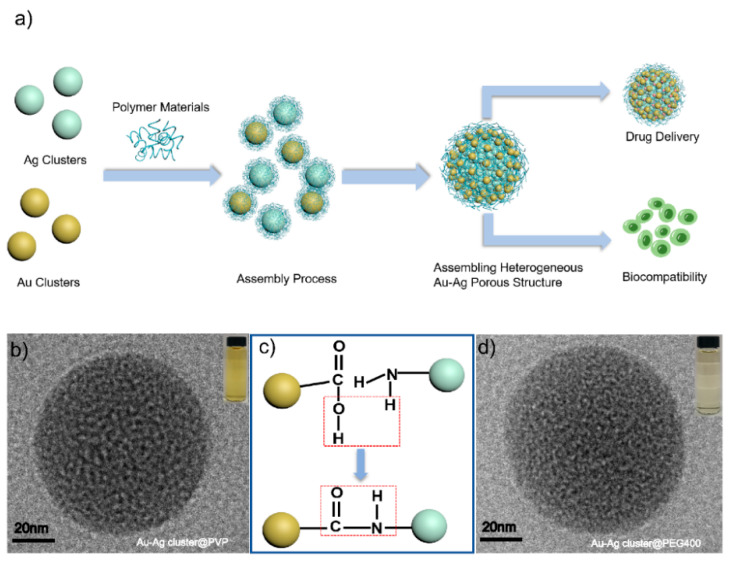
(**a**) Schematic diagram of formation and synthesis of heterogeneous assembly structures of precious metal NCs (Au, Ag); (**b**)HR-TEM images of Au-Ag cluster @PVP; (**c**) Self-assembled mechanism of Au-Ag NCs and (**d**) HR-TEM images of Au-Ag cluster @PEG400.

**Figure 2 ijms-23-11197-f002:**
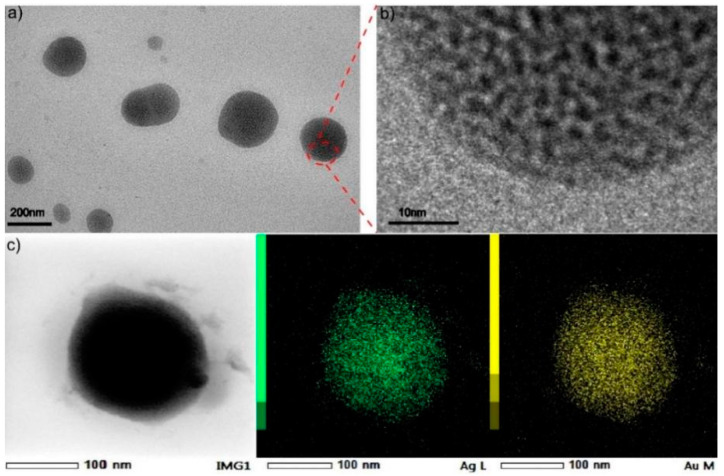
(**a**)TEM images of Au@ Ag NCs by using PVP as soft template; (**b**)HR-TEM images and (**c**) the elemental mapping images of Au, Ag in assembled Au@ Ag NCs.

**Figure 3 ijms-23-11197-f003:**
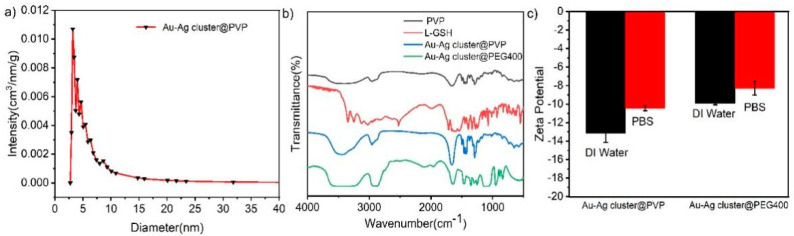
(**a**) Pore size distribution curves of assembled heterogeneous Au-Ag NCs; (**b**) Powder FT-IR spectra of Au-Ag cluster @PVP and Au-Ag cluster @PEG400, respectively; (**c**) Zeta potential values of Au-Ag cluster @PVP and Au-Ag cluster @PEG400, which were incubated in PBS and water medium, respectively.

**Figure 4 ijms-23-11197-f004:**
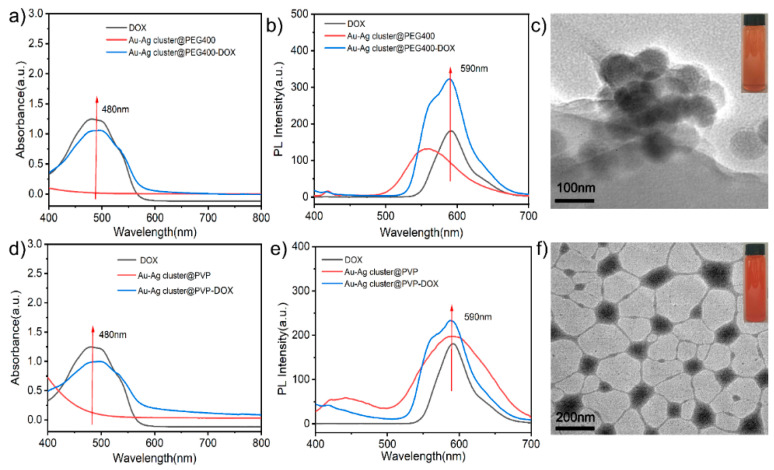
(**a**) UV-vis absorption and (**b**) PL spectra of DOX, Au-Ag cluster@PEG400 and Au-Ag cluster@PEG400-DOX, respectively; (**c**)TEM images of Au-Ag cluster@PEG400-DOX; (**d**) UV-vis absorption and (**e**) PL spectra of DOX, Au-Ag cluster @PVP and Au-Ag cluster @PVP-DOX, respectively; (**f**) TEM images of Au-Ag cluster@PEG400-DOX.

**Figure 5 ijms-23-11197-f005:**
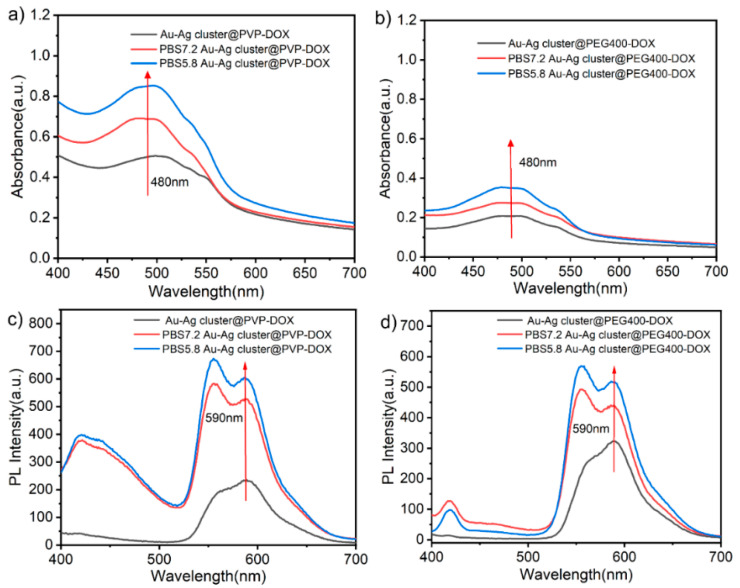
(**a**,**b**) UV-vis absorption of Au-Ag cluster @PVP-DOX, Au-Ag cluster @PEG400-DOX and it in PBS buffer at different pH value; (**c**,**d**) PL spectra of Au-Ag cluster @PVP-DOX, Au-Ag cluster @PEG400-DOX and it in PBS buffer at different PH.

**Figure 6 ijms-23-11197-f006:**
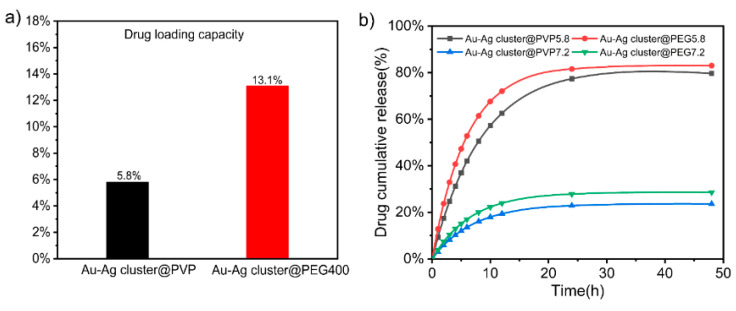
(**a**) Drug loading capacity of Au-Ag cluster @PVP and Au-Ag cluster@PEG400, respectively; (**b**) Time-dependent of DOX releasing profiles of Au-Ag cluster @PVP-DOX and Au-Ag cluster @PEG400-DOX at pH = 5.8 and 7.2, respectively.

**Figure 7 ijms-23-11197-f007:**
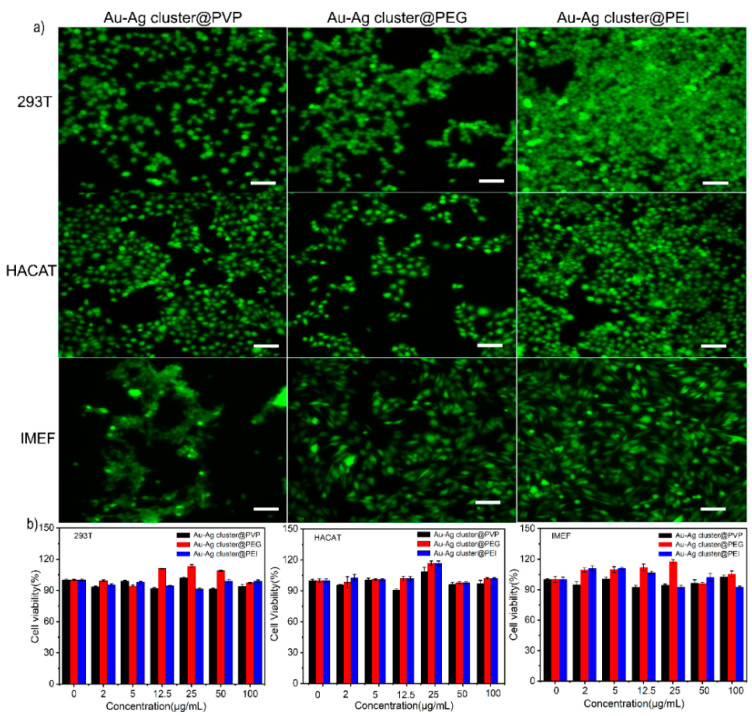
(**a**) Representative fluorescence microscopy images of 293T, HACAT and IMEF cells, which were treated by using 50 µg mL^−1^ colloidal solutions (Au-Ag cluster @PVP, Au-Ag cluster@PEG400 and Au-Ag cluster @PEI) after 2 days, in preparation for biocompatibility measurements. The scale bar in the inset corresponds to 100 µm; (**b**) Relative cell viability of different cell lines 293T, HACAT and IMEF treated with different concentrations of modified Au-Ag cluster @PVP, Au-Ag cluster @PEG400 and Au-Ag cluster @PEI for 2 days.

**Figure 8 ijms-23-11197-f008:**
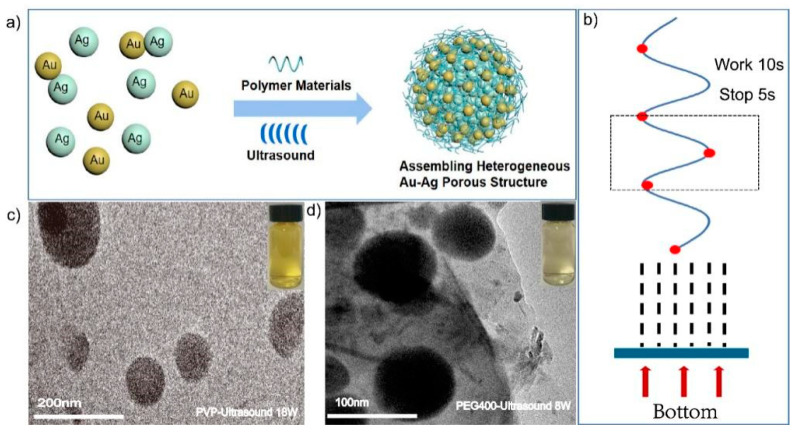
(**a**) Schematic diagram of formation and synthesis of heterogeneous assembly structures of precious metal NCs (Au, Ag) under ultrasonic conditions; (**b**) Simulation model while applying ultrasound fields (6 MHz, 8 W, 13 W, 18 W) from bottom incentive (ultrasonic emission 10 s, stop 5 s); (**c**) TEM images of Au-Ag cluster @PVP prepared at 18 W ultrasonic power; (**d**) TEM images of Au-Ag cluster @PEG400 prepared at 8 W ultrasonic power.

## Data Availability

The data presented in this study are available on request from the corresponding author.

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
