# Peer review of "Metal Cluster Triggered-Assembling Heterogeneous Au-Ag Nanoclusters with Highly Loading Performance and Biocompatible Capability"

_ijms, 2022, doi:10.3390/ijms231911197_

Round 1
Reviewer 1 Report
In this work the Authors proposed the use of Au-Ag clusters as drug delivery. The topic is of great interest and the research aim is clear. However several points throughout the manuscript need some improvements. As a general comment, the English structures of the manuscript should be reviewed. For instance, the adverb “in which” is inappropriately used both in the abstract and throughout the introduction.
Detailed list of comments and revision requests are reported below.
- Lines 199-219: The discussion about S1-S3 figures is not totally clear. What is the difference between the TEM images of Au-Ag cluster @PVP in Fig. S1 and S3? It seems that their morphology is pretty different. In addition, the polyethyleneimine is mentioned in the text but there are no reported figures about its use as stabilizer.
- Lines 224-225: Here the Au-Ag cluster @PEI are mentioned. However, other stabilizers are reported in the caption of S4 figure (Au-Ag cluster@PEG400/PVP-DOX). The Authors should clarify this point.
- Lines 253-255: “The polymer materials (PVP, PEG400 and PEI) could keep the stability while individual metal cluster integration, it maintained the dispersion and non-agglomerative status as much as possible.” The structure of this sentence should be revised.
- Lines 257-258: “In all this, motivation is considered a spontaneous work process.” What is the meaning of “motivation” in this sentence?
- Fig. 3 and lines 264-295: there are different points in the IR discussion that should be improved.
- First of all, why were two different IR setups used (FT-IR and ATR-FT-IR)? It would be more appreciable to use the same methodology.
- In the Fig. 3b the spectrum of Au-Ag cluster@PVP seems comparable to the one of only PVP, suggesting just that the presence of PVP as stabilizer is significant. The Authors should comment on this aspect.
- The spectrum of the Au-Ag cluster @PEG400 in Fig. 3c is totally coincident with the IR spectrum of water (reference: NIST Chemical Book) so it is useless to characterize the Au-Ag clusters. This is to be expected: the water in which the Au-Ag clusters are dispersed is more abundant than the Au-Ag clusters themselves, so the ATR laser beam predominantly interacts with the solvent. The IR measurements should be performed using a KBr tablet in typical FT-IR setup.
- Caption Fig. 3: “(d) Zeta potential curves of Au-Ag cluster @PVP and Au-Ag cluster @PEG400, which were incubated in PBS and water medium, respectively”. The mentioned graph in Fig. 3d is not a curve but an histogram, so “zeta potential curves” should be replaced with “zeta potential values”.
- Is not totally clear the values of the zeta potential of the nanostructures. It seems that they are in the range of 2-7 mV. However, in this range the nanoparticles aren’t stable, contrary to the Authors discussion in the lines 290-291. In general, the nanoparticles that possess zeta potentials of more than +30 mV or less than –30 mV are considered as a stable colloidal suspension system (e.g. see: 10.1016/B978-0-12-817909-3.00010-8). In addition, the figure S5d is mentioned in the line 290 as figure S4d.
- Fig. 6: It seems that the different drug loading capacity of clusters with different stabilizers doesn’t significantly affect drug release overtime, which is comparable for both PVP and PEG clusters. Have the Authors an idea of this behaviour?
- Lines 496-497: “Supporting Information: Additional information as noted in the text. This material is available free 496 of charge via the Internet at http://pubs.acs.org.” Why was another scientific journal mentioned if the Authors submitted their manuscript for an MDPI journal?
In conclusion, major revisions are necessary for paper publication.
Reviewer 2 Report
The paper presents an approach designed for making heterogeneous Au-Ag nanoclusters. The formed structures exhibit relative structural stability, porous characteristic, and pH-dependent properties. The drug loading and releasing performance are tested in connection to the structural featurs of the nanoclusters. It proved that this heterogeneous structure is biocompatible and that the particle size and structure of assembled heterogeneous Au-Ag nanoclusters were greatly affected by the ultrasound power, while the formation time of assembled structures was shortened at the same time. It was also proven that the physical energy release, by acoustic cavitation, might induce the assembling process through influencing the internal bond of polymers. The paper cannot be published in its present form, in spite of the clear relevance of the results obtained, because it is so poorly written in English, to result hardly readable. Here a just a few examples, taken from the Abstract:
GSH-protected -> glutathione-protected
exhibit the relative biocompatibility was enhanced than before.
-> exhibit that the relative biocompatibility was enhanced.
In all, there is one novel approach is exhibited, ->
In all, a novel approach is exhibited,
Now, moving to the Introduction:
referees, it bases -> references, based
"Among combined works, metal cluster could remain the more activity and exhibit particularity in tested physicochemical works. "Here the Auhtors probably mean something like: "Amongst composite materials, metal cluster attracted the most attention, as they exhibited special features, when their properties were tested in physicochemical works." But it is mere guesswork on my side. The meaning of the sentence is obscure.
few atoms with a higher surface area WHAT IS THE SURFACE AREA OF ATOMS? They mean nanoclusters surface area? Also, the nanoclusters mentioned in lines 52-53 are made of several hundred atoms, and not just a few atoms. Please explain.
controlling metal atoms quality HOW DO YOU CONTROL THE QUALITY OF ATOMS? They probably mean: controlling the cluster chemical purity
In addition, the noble metal clusters have good fluorescence property26-29 due to the spatial confinement of free electrons, it leads to discrete electron hopping 30, 31. In which the electron energy levels disperse from continuous to discontinuous states. The electron hopping becomes active and result strong light absorption32
->
In addition, the noble metal clusters have good fluorescence property26-29 due to the spatial confinement of free electrons, which leads to discrete electron hopping 30, 31, with the electron energy levels dispersing from continuous to discontinuous states. The electron hopping becomes active and results in strong light absorption32
As this is becoming to be real tedious, I now skip to the Conclusion. Line 475 drawbacks of in preparing -> drawbacks in preparing
Line 479: reaction, it could -> reaction. This could
Line 484: building block -> building blocks
Line 485: assembled works -> assembling procedure
Line 485: seems that -> seems to indicate that
Line 489: It proved -> It proved that
Line 490 (acoustic cavitation)-> (by acoustic cavitation)
Line 491: through influence -> through influencing
In summary, this paper is poorly written, although the results may be interesting. It is necessary that the Authors make an effort to rewrite the paper more carefully, not only from the English language point of view, but also from that of the consistency of the sentences, in order to enhance the readability of the manuscript, thus bringing it to the standards of scientific literature.
Round 2
Reviewer 1 Report
Dear Authors,
although the English structure of the manuscript is improved, some comments were not taken into account or the Authors' responses are not clear. In particular, the comments regarding the ATR-FT-IR spectra of Au-Ag cluster @PEG400 (Fig. 3c) and the zeta potential (Fig. 3d) are unacceptable and unclear. Those figures cannot be published in that way. As it was explained in the first report, the IR spectrum in Fig. 3c basically shows the main moieties of water molecules. In addition, no changes are reported in the zeta potential discussion (although required) and the meaning of the Fig. 3d is not totally clear.
In conclusion, major revisions are necessary for paper publication.
Round 3
Reviewer 1 Report
The Authors took into account the suggestion and comments of Reviewers. They revised the manuscript and improved its quality. In conclusion, the paper is now suitable for publication.
Author Response
Comments and Suggestions for Authors:
The Authors took into account the suggestion and comments of Reviewers. They revised the manuscript and improved its quality. In conclusion, the paper is now suitable for publication.
--Thanks so much for reviewer’s work. Really appreciate for this.